# Indicators of high-quality general practice to achieve Quality Equity and Systems Transformation in Primary Health Care (QUEST-PHC) in Australia: a Delphi consensus study

Phyllis Lau[1,2]*, Samantha Ryan[1,2], Baneen Alrubayi[1], Lucy Bannister[1], Dylan Pakkiam[1], Penelope Abbott[1,2], Kathy Tannous[2,3], Steven Trankle[1,2], Kath Peters[2,4], Andrew Page[2], Natalie Cochrane[1,2], Tim Usherwood[5,6], Jennifer Reath[1,2]

1 School of Medicine, Western Sydney University, Sydney, New South Wales, Australia, 2 Translational Health Research Institute, Western Sydney University, Sydney, New South Wales, Australia, 3 School of Business, Western Sydney University, Sydney, New South Wales, Australia, 4 School of Nursing and Midwifery, Western Sydney University, Sydney, New South Wales, Australia, 5 Faculty of Medicine and Health, The University of Sydney, Sydney, New South Wales, Australia, 6 George Institute for Global Health, Sydney, New South Wales, Australia

* Phyllis.lau@westernsydney.edu.au

## Abstract

### Objectives

This study aimed to achieve wider consensus on the relevance and feasibility of the Quality Equity and Systems Transformation in Primary Health Care (QUEST-PHC) indicators and measures developed for Australian general practice.

### Methods

Partnering with eight Primary Health Networks (PHNs) across four states, we conducted a Delphi consensus study consisting of three rounds of online survey with general practice experts including general practitioners, practice nurses and PHN staff members. Participants rated each measure for relevance and feasibility, and provided input into the implementation of a quality indicator tool. Each measure required ≥70% agreement in both relevance and feasibility to achieve consensus. Aggregated ratings were statistically analysed for response rates, means, standard deviations, ranges, and level of agreement. Sub-group analyses were conducted to compare the aggregated ratings between practice and PHN staff, and between clinicians and non-clinicians in the practice staff. Qualitative responses were analysed thematically using an inductive approach.

**Data availability statement:** All relevant data are within the paper and Supporting Information files.

**Funding:** This study was funded by the Digital Health Cooperative Research Centre https://www.digitalhealthcrc.com/. The funder had no role in study design, data collection and analysis, decision to publish, or preparation of the manuscript.

**Competing interests:** The authors have declared that no competing interests exist.

## Results

Ninety-four participants participated in Round 1 survey; 61 completed all three rounds. All measures reached the consensus threshold for both relevance and feasibility; 19 were slightly less feasible when compared with other measures. Although in general the participants scored similarly and their agreements were statistically significant, subgroup analyses showed that PHN staff scored feasibility of some measures slightly lower than practice staff (e.g., patients screened for adverse childhood experiences), and clinicians also scored the feasibility of some measures slightly lower than non-clinicians (e.g., patient perceptions of preventative health discussion on unsafe sexual practices).

## Conclusions

The QUEST PHC suite of indicators and measures have reached consensus in this Delphi study. Whilst the feasibility of some measures still needs considerations, the QUEST PHC suite provides a framework for defining and measuring high-quality general practice to enable reporting to inform quality improvement and alternative funding models for Australian general practice.

## Introduction

High-quality primary health care (PHC) is key to addressing health needs, containing spiralling health costs and providing equitable community-based care Section 1. Quality PHC is defined by the World Health Organisation (WHO) as a "whole-of-society approach to effectively organise and strengthen national health systems to bring services for health and wellbeing closer to communities" with three key components: "comprehensive integrated health services that embrace primary care as well as public health goods and functions as central pieces; multi-sectoral policies and actions to address the upstream and wider determinants of health; and engaging and empowering individuals, families, and communities for increased social participation and enhanced self-care and self-reliance in health" [1,2].

The Quadruple Aim is a framework often used to guide and evaluate primary health systems, in Australia and around the world [3,4]. It states that effective health-care improvement must take into account the care experience of individual patients, the health of populations, health care costs and the wellbeing of health care providers [4–8]. The Royal Australian College of General Practitioners promotes a model of high-performing patient care in general practice based on addressing the four principles of the Quadruple Aim to achieve a sustainable healthcare system [3]. Although the Quadruple Aim was gradually expanded in 2022 to the Quintuple Aim to include a fifth aim of advancing health equity, the latter remains relevant in Australia [4,8]. Australian research shows that an integrated and patient-centred health care system that is properly funded is key to ensuring high-quality care [9,10]. The challenge lies in ascertaining agreement on what constitutes high-quality PHC in Australia to guide an appropriate funding model.

## Why are high-quality care indicators needed?

A general practice indicator is "a measurable element of practice performance for which there is evidence or consensus that it can be used to assess the quality, and hence change in the quality, of care provided" [11]. Quality indicators offer insight into service quality and trends, and highlight potential areas for review of healthcare decision making, quality improvement and further research [12].

General practice indicators are critical to informing quality improvement and supporting high-quality primary health. Many countries collect PHC data such as health outcomes data and data related to utilisation of health services, like the Medicare Benefits Scheme and Pharmaceutical Benefits Scheme datasets in Australia [13,14]. and the Nivel's Primary Care Database in The Netherlands [15]; and also for research purpose such as the Clinical Practice Research Datalink (CPRD) in the UK to inform quality improvement including drug safety, use of medicines, health care delivery and disease risk factors [16].

Standardized and evidence-based indicators to measure and track high quality clinical performance and outcomes in general practice are necessary for the profession's accountability and to identify population needs and gaps in patient care [17, 18].

In Australia, general practice data is routinely collected through the Practice Incentives Program Quality Improvement initiative (PIP QI), launched in August 2019 by the Australian Government to improve patient outcomes and to provide incentive payments to general practices contributing to this initiative [19–22]. Primary Health Networks (PHNs)., established in 2015 across Australia to support PHC, in addition to collecting PiP QI data from general practices, also extract data to inform local practice quality improvement [21]. However, there is a lack of consistency across the PHNs in data content, quality of the data collected and, importantly, in the quality improvement outcomes achieved through this process [23].

The Australian general practice services are also funded through fee-for-service (FFS) payments via the universal health coverage system, Medicare, supplemented in some cases by patient contributions. This funding model rewards service throughput rather than quality [24,25]. Government funding of PHC in Australia is based to a larger extent on FFS payments than in other developed countries [26].

There is a need in Australia to establish evidenced-based indicators of high-quality general practice to drive quality of care and to inform general practice funding reform that provides a greater incentive for high quality care. The development and implementation of quality indicators are complex and require evidence, practical experience, literature review, consultation and verification with experts and stakeholders [12].

## QUEST PHC indicators and measures

In 2019–2020, researchers at Western Sydney University (WSU) identified a suite of 79 evidence-based indicators and their corresponding 128 measures of high-quality general practice based on the literature and extensive consultation with PHNs in western Sydney [25]. Key literature was analysed to identify the attributes of high-quality general practice in order to construct a framework for the indicators and measures. The attributes align with the elements of the Quadruple Aim and are expressed as 'accountabilities': highlighting accountability to our patients, professions, community and society [5–7,25] The indicators and measures are further grouped under structures (S), processes (P) and outcomes (O) according to a Donabedian framework. In this framework 'structures' describe the context in which care is delivered, e.g., buildings, staff, financing and equipment; 'processes' denote what is actually done in giving and receiving care; and 'outcomes' refer to the effects of healthcare on patients and populations [25,27,28].

The overall aim of the QUEST PHC project is to develop Australia's first comprehensive, evidence-based, professionally endorsed tool for analysing and reporting data across all components of high-quality general practice in Australia, thereby informing quality improvement and potentially providing a framework for alternative funding models. This paper reports on the results of the Delphi consensus study conducted, as part of the content validation process, to establish

wider consensus with general practice and PHN staff across four states on the relevance and feasibility of the identified suite of indicators and measures for the Australian context [23].

## Methods

The protocol of this Delphi consensus study including the justification of the methodology has been published previously [23]. A summary of this and variations to the original protocol, including justifications for these, are described in this section.

### Ethics approval

This research had ethics approval from Western Sydney University Human Research Ethics Committee (ID H14460). All participants provided written consent before participation.

### Study design

This Delphi study used a concurrent mixed-methods three-round survey to seek consensus across an expert group of general practice and PHN staff involved in quality improvement initiatives [29,30], on the relevance and feasibility of the 79 indicators and their associated 128 measures previously developed through literature review and stakeholder consultation (S1 File) [23,25]. The guidelines for Conducting and REporting of DElphi Studies (CREDES) was used to guide the reporting of this study [31].

### Project governance

The QUEST PHC project was overseen by the Project Control Group that consisted of representatives from the Digital Health Cooperative Research Centre (DHCRC) that funded the research, and eight primary health organisations: Brisbane North PHN, Central and Eastern Sydney PHN, Nepean Blue Mountains PHN, North Western Melbourne PHN, South Western Sydney PHN, Western Sydney PHN (WentWest), Western Australia Primary Health Alliance, and Western NSW PHN. In addition, the research had a steering committee that consisted of the representatives of the PHNs noted above, the Royal Australian College of General Practitioners (RACGP), Australian College of Rural and Remote Medicine (ACRRM), Justice Health NSW (New South Wales) and SA (South Australia) Prison Health Service. It provided strategic direction and advice to the research team on dissemination and collaboration with relevant stakeholder groups.

### Setting

The Delphi study was undertaken in four states (New South Wales, Queensland, Victoria and Western Australia) in Australia across regions of the eight PHNs, covering a total area of 2,942,817km$^2$ in metropolitan and rural Australia, and a diverse population of over 9.6 million with over 3,000 general practices. The characteristics of the PHNs, their geographical locations and the populations in their regions are summarised in the protocol paper previously published [23].

### Sample Size

Formal sample size calculations are not required in the Delphi methodology. The median number of participants reported for content validation in previous Delphi studies involving the selection of healthcare quality indicators was 17 [32]. Following consultation with the Project Governance Group and considering the participation of a general practice and primary health care expert group rating on 128 measures, particularly during the COVID-19 pandemic, the study therefore adopted a pragmatic approach and aimed to recruit a minimum of 80 participants for Round 1, with an anticipated retention rate of 40–45% in subsequent rounds to attempt to meet the minimum sample size requirement.

## Participants and recruitment

Purposive and convenience sampling was used to recruit participants that included GPs, practice nurses, practice managers and key PHN staff familiar with quality improvement initiatives in Australian general practice. Information packs including an email template, participant information sheet and participant consent form were provided to PHNs to distribute to their nominated staff members and general practices in their regions. Recruitment to the Delphi panel commenced on 26th October 2021. Each practice recruited was asked to nominate one to two practice staff to participate in the survey.

## Criteria for the Delphi participants to consider

Participants assessed the indicators and measures, grouped under the attributes of high-quality general practice framework and mapped against the Quadruple Aim (Table 1) [5–7]. S1 File provides the full list of the QUEST PHC indicators and measures.

## Data collection

Three surveys were constructed using the online survey platform Qualtrics (Qualtrics, Provo, UT, USA. https://www.qualtrics.com). They were pilot tested with three academic colleagues at the Western Sydney University Department of General Practice and the research team, and revised to improve comprehensibility and the functioning of the survey.

Unique links were emailed to participants on the morning that each round opened. Each round was estimated to take 20–30 minutes to complete. All survey participants were anonymised to their PHNs and other participants. A password protected file was maintained by the research team with participants' identifying information.

The recruitment, retention and data collection processes were heavily impacted by the NSW 2021 floods [33], the COVID-19 Delta outbreak [34] and the vaccination roll-out throughout Australia [35]. General practices and PHNs were directly involved in disaster and emergency responses, and pandemic prevention and control. The original protocol was to open each round of survey for three weeks, remind participants up to three times via email, and analyse the results over two weeks in between rounds, however, the duration of each round required substantial extension to accommodate these disasters and the pandemic response.

**Table 1. High-quality general practice attribute framework, alignment with the Quadruple Aim and number of indicators and measures under each attribute [23].**

| Attribute | Definition [25] | Aligning with Quadruple Aim [5] | Number of indicators and measures |
|---|---|---|---|
| Attribute One: Accountability to our patients | High-quality general practice provides evidence-based, person-centred and comprehensive care (including preventive, chronic and acute care), with patient-general practice team partnerships as a key aim. | Improving the individual experience of care | 47 indicators with 79 measures |
| Attribute Two: Professionally accountable | High-quality general practice is:<br>• high-functioning multidisciplinary teams engage in continuing care that is coordinated and integrated with other services and the medical neighbourhood;<br>• supported by clinical governance, staff training and data-enabled practice quality improvement;<br>• engaged with general practice education and/or research to provide a means of sustaining the quality of the health system. | Improving the work life of clinicians and staff | 19 indicators with 31 measures |
| Attribute Three: Accountable to the community | High-quality general practice is accessible, responsive to population health needs and focussed on providing equitable care. | Improving the health of populations | 10 indicators with 16 measures |
| Attribute Four: Accountable to society | High-quality general practice promotes efficient stewardship of health resources. | Reducing the per capita costs of care for populations | 2 indicators with 2 measures |

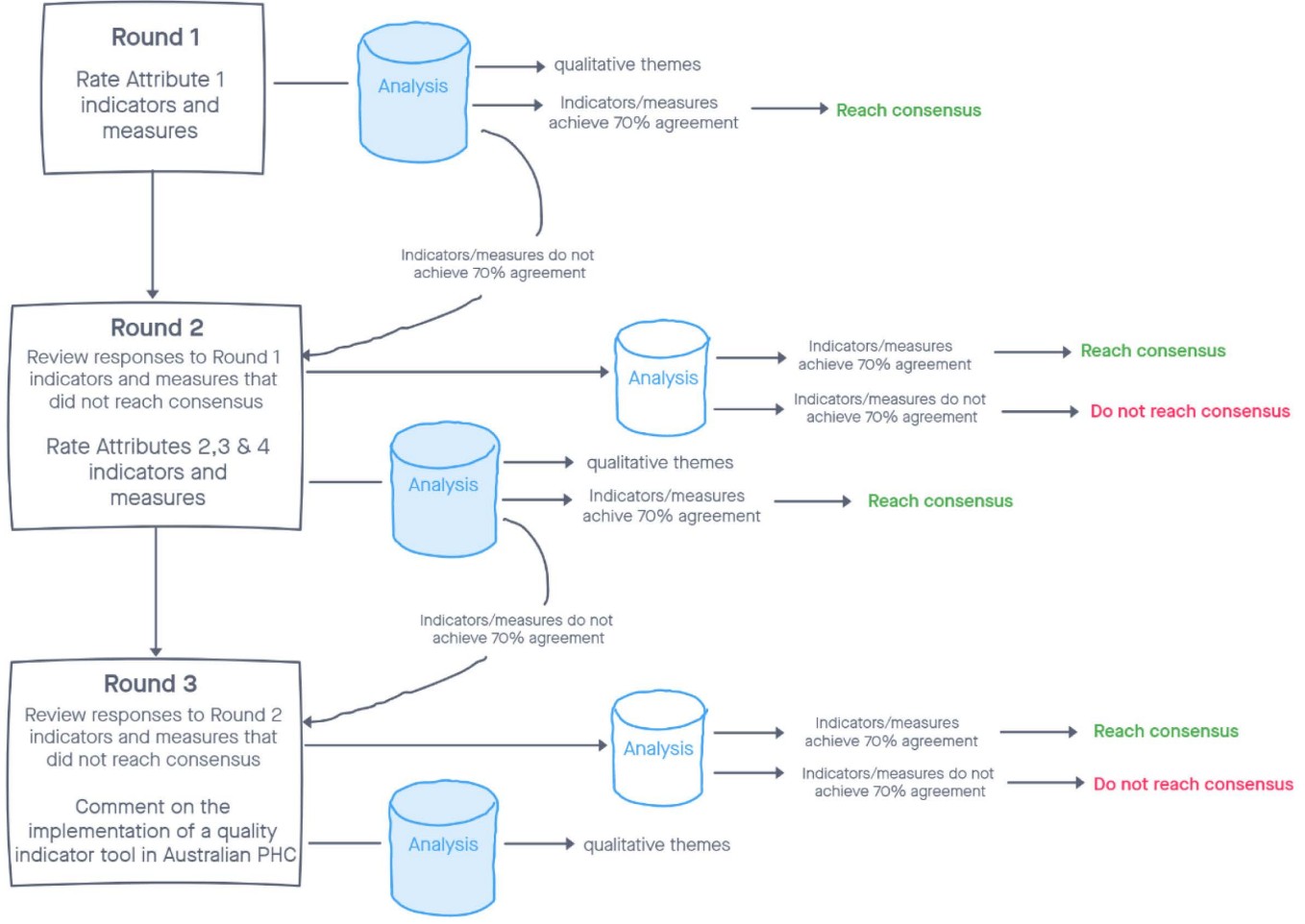

**Fig 1. Delphi study rating process.**

**Rating process.** The flow of the Delphi study rating process is shown in Fig 1.

Participants were asked to rate each indicator and measure for relevance and feasibility on a 4-point Likert scale (1 = irrelevant/infeasible, 2 = somewhat irrelevant/infeasible, 3 = somewhat relevant/feasible, 4 = relevant/feasible.) Relevance is defined as the value and appropriateness of an indicator/measure in Australian general practice; feasibility is defined as the applicability and implementability of an indicator/measure in Australian general practice. Participants were not required to rate every measure [23].

The levels of consensus in the Delphi methodology vary depending on size of the expert panel and the aim of the research [36]. The threshold for consensus for this study was defined 'a priori' based on previous research experience [37]. The mean score of each measure was required to reach a minimum of 70% agreement (i.e., combined scores of 3 and 4 on the Likert scale) in both relevance and feasibility to achieve consensus. We determined that this was a pragmatic and reasonable approach for establishing consensus in view of the size of the expert panel, the aim of the research, and the diverse and complex general practice settings [23,36].

**The survey. Round 1:** Round 1 commenced on 26th November 2021, remained opened, following three extensions to increase participant recruitment, for 17 weeks, and closed on 24th March 2022. Participants provided demographic

information including name, age, gender, job position, and number of years of experience in their role. Indicators and measures under attribute one (S1 File and Table 1) were presented to participants for rating. Qualitative open-ended items were included at the end of each topic area for participant comments, including recommendations for additional indicators or measures.

**Round 2:** Round 2 commenced 18th February 2022. To accommodate for the extensions of Round 1 and to allow later-recruited participants time to complete Rounds 1 and 2, Round 2 remained opened for eight and a half weeks and closed on 19th April 2023.

In this round, indicators and measures under attributes two, three and four (S1 File and Table 1) were presented to participants for rating and comments, as before in Round 1, for each subgroup of indicators. Qualitative open-ended items were again included at the end of each topic area for participant comments and recommendations for additional indicators or measures.

**Round 3:** Round 3 commenced on 22nd April 2022, opened for five weeks and closed on 27th May 2022. In this round, participants re-rated items that did not reach consensus in previous rounds. They were then asked 22 open-ended questions about their overall views on the proposed indicators and measures and whether they reflected high-quality care in general practice; benefits and challenges in implementation at the general practice, PHN, and national levels; patient-reported measures (or PRMs that capture information via surveys, which ask patients about their healthcare experiences and the outcomes of their care [38]); and the Delphi process.

## Quantitative analysis

Participants' demographics were analysed descriptively using SPSS® (IBM version 26.0, 2020).

Scores for each measure were dichotomised by combining scores of 1 and 2 as 'irrelevant or infeasible', and scores of 3 and 4 as 'relevant or feasible'. Our protocol stipulated that measures that achieved ≥70% in relevance but not feasibility were to be included in a 'blue skies' category for future consideration, and those that achieved ≥70% in feasibility but not relevance were to be discarded. Proportions of ratings 3 (somewhat relevant/feasible) and 4 (relevant/feasible) was also calculated to determine the strength of the consensus on each measure.

The aggregate results of participants' ratings of each measure were analysed for percentage response rates for each score, means, medians, ranges, modes, skewness and level of agreement. The analysis in the original protocol also included interquartile ranges and associated group rankings, but due to the skewness of the scores, these statistics and standard deviations or median ranges did not provide added value, so instead ranges and skewness were calculated to demonstrate the distribution of the data.

Two sub-group analyses were performed between general practice staff and PHN staff, and between clinicians and non-clinicians within general practice staff. Aggregate results of each subgroup participants' ratings of each measure were analysed as above.

For each round and outcome (relevance and feasibility), the agreement for each pair of ratings was also calculated as the percentage of equal ratings. The agreement was not calculated for pairs of ratings that did not rate at least 90% of all items to improve the quality of the assessment. Independent t-tests were used to compare the mean agreement between subgroups. The use of ANOVA models was not possible due to the dependencies between groups. All analyses were performed using R, version 4.2.2 [39].

## Qualitative analysis

Reflexive thematic analysis, involving searching for meaningful patterns within datasets, was utilised to analyse the open-ended survey responses [40]. An inductive approach was undertaken in which themes identified were data driven rather than being fitted into an existing coding frame. The data from each round was coded by four researchers separately in Microsoft word. The first step involved familiarisation, in which three researchers read the data from each round. Initial

codes were used to form the basis of a coding frame, which was subsequently developed, tested, and refined through discussions with all four researchers. The coded data was then re-examined and relationships across codes were mapped. Overarching themes and subthemes were placed into a table with the corresponding data which was further reviewed and refined by the research team.

Although the original protocol stated that codes would be grouped according to the accountability attributes, the nature of the qualitative feedback did not elicit patterns that were suitable for this approach [23].

## Results

A total of 94 participants were recruited for Round 1 of the survey, including 68 practice staff (from 57 general practices) and 26 PHN staff. Thirty-three practice staff were clinicians and 32 were non-clinicians. Three did not specify and were excluded from the overall analysis. Table 2 summarises the participant demographic, and Table 3 outlines participant retention across the three rounds. Sixty-one participants completed all three rounds (retention rate 64.9%).

### Quantitative results

Fig 2 shows the distribution of the means and standard deviations of scores related to the 128 measures. Most of the mean scores for relevance and feasibility were on the higher ends of the Likert rating scale. The mean plus one standard deviation (SD) for most of the measures are higher than the top score of 4 because the data is not normally distributed and there is a strong left skewness. Detailed data is available in S2 File.

**Consensus across measures in the four attributes.** Table 4 shows the overall rating data across all four attributes. All measures, except one, achieved ≥70% agreement for both relevance and feasibility in the first rating. Measure P7

**Table 2. Participant demographics.**

| Demographics | | Practice staff (n = 68) | PHN staff (n = 26) | Overall (n = 94) |
|---|---|---|---|---|
| Mean age [years] | | 47.7 | 43.3 | 46.5 |
| Gender [number (%)] | Female | 49 (72.1%) | 25 (96.2%) | 74 (78.7%) |
| | Male | 19 (27.9%) | 1 (3.8%) | 20 (21.3%) |
| Average Professional experience [years] | | 17.9 | 18.6 | 18.1 |
| Clinicians [number (%)] | | 33* (48.5%) | | 33 (35.1%) |
| General practitioners | | 19 (27.9%) | | 19 (20.2%) |
| Practice/registered nurse | | 14 (20.5%) | | 14 (14.9%) |
| Non-clinicians [number (%)] | | 33* (48.5%) | | 33 (35.1%) |
| Practice manager | | 30 (44.1%) | | 30 (31.9%) |
| Nurse manager | | 2 (2.9%) | | 2 (2.1%) |
| Associate Director | | 1 (1.5%) | | 1 (1.1%) |
| Manager | | | 7 (26.9%) | 7 (7.4%) |
| Quality improvement | | | 7 (26.9%) | 7 (7.4%) |
| Support officer | | | 3 (11.5%) | 3 (3.2%) |
| Practice/Nurse facilitator | | | 4 (15.4%) | 4 (4.3%) |
| Team leader | | | 2 (7.7%) | 2 (2.1%) |
| Health data officer | | | 1 (3.8%) | 1 (1.1%) |
| Immunisation and screening coordinator | | | 1 (3.8%) | 1 (1.1%) |
| Practice liaison/support officer | | | 1 (3.8%) | 1 (1.1%) |

*Two practice staff did not specify whether they were clinicians or non-clinicians.

**Table 3. Retention of participants over the three rounds of survey.**

| Practice staff | | | | PHN staff | | | |
|---|---|---|---|---|---|---|---|
| Round 1 | Round 2 | Round 3 | % Completion (b/a) | Round 1 | Round 2 | Round 3 | % Completion (d/c) |
| 68[a] | 46 | 38[b] | 55.9% | 26[c] | 26 | 23[d] | 88.5% |

Total participants completed all three rounds = 61 (RR 64.9%)

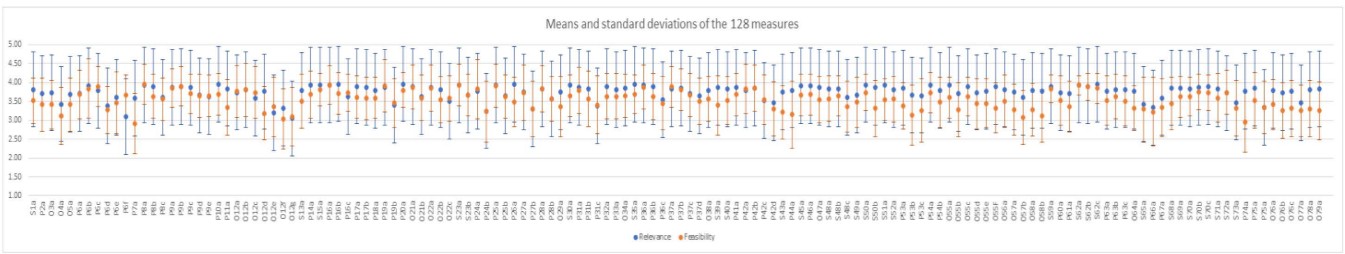

**Fig 2. Distribution of the means and standard deviations of scores for each QUEST PHC measure.**

**Table 4. Overall ratings of the indicators and measures.**

| | Attribute 1 (47 indicators with 79 measures) | | Attribute 2 (19 indicators with 31 measures) | | Attribute 3 (10 indicators with 16 measures) | | Attribute 4 (2 indicators with 2 measures) | |
|---|---|---|---|---|---|---|---|---|
| | Relevance | Feasibility | Relevance | Feasibility | Relevance | Feasibility | Relevance | Feasibility |
| No. of raters | 80-94 | 80-94 | 72-73 | 72-73 | 72 | 72 | 72 | 72 |
| No. of measures reached consensus after 1st rating | 79 of 79 | 78[a] of 79 | 31 of 31 | 31 of 31 | 10 of 10 | 10 of 10 | 2 of 2 | 2 of 2 |
| No. of measures reached consensus after 2nd rating | N/A | 1 of 1 | N/A | N/A | N/A | N/A | N/A | N/A |
| Mean score ± standard deviation | 3.7±0.2 | 3.6±0.2 | 3.8±0.1 | 3.5±0.2 | 3.7±0.1 | 3.5±0.2 | 3.8±0.0 | 3.3±0.0 |
| Mean score range | 3.0 to 4.0 | 2.9 to 4.0 | 3.3 to 4.0 | 3.1 to 3.9 | 3.5 to 3.9 | 3.0 to 3.8 | 3.8 to 3.8 | 3.3 to 3.3 |
| Median scores | 4 | 3 and 4 | 4 | 3 and 4 | 4 | 3 and 4 | 4 | 3 |
| Mode scores | 4 | 3 and 4 | 4 | 3 and 4 | 4 | 3 and 4 | 4 | 3 and 4 |
| Skewness range | −6.3 to −1.1 | −2.4 to −0.3 | −5.7 to −1.1 | −3.5 to −0.3 | −3.7 to −0.9 | −2.6 to −0.4 | −3.8 to −2.7 | −0.9 to −0.8 |
| Score 1 (% range) | 0.0 to 2.1 | 0.0 to 5.0 | 0.0 to 8.3 | 0.0 to 6.9 | 0.0 to 1.4 | 0.0 to 4.2 | 0.0 to 1.4 | 1.4 to 2.8 |
| Score 2 (% range) | 0.0 to 8.5 | 0.0 to 25.6 | 0.0 to 6.9 | 0.0 to 21.9 | 0.0 to 9.7 | 1.4 to 22.2 | 0.0 to 2.8 | 11.1 to 12.5 |
| Score 3 (% range) | 2.3 to 34.0 | 12.0 to 53.4 | 2.7 to 33.3 | 6.9 to 54.8 | 9.7 to 34.7 | 16.7 to 51.4 | 12.5 to 12.5 | 40.3 to 44.4 |
| Score 4 (% range) | 55.3 to 96.6 | 24.4 to 88.0 | 55.6 to 95.9 | 27.4 to 93.1 | 55.6 to 88.9 | 26.4 to 79.2 | 84.7 to 86.1 | 41.7 to 45.8 |
| Combined scores 3 and 4 (% range) Proportion that is rating 3 (% range) Proportion that is rating 4 (% range) | 89.3 to 100.0 2.3 to 38.1 61.9 to 97.7 | 71.1 to 100.0 12.0 to 66.2 33.8 to 88.0 | 85.9 to 100.0 2.8 to 37.0 63.2 to 97.3 | 76.4 to 100.0 6.9 to 66.3 34.0 to 94.0 | 90.1 to 100.0 10.0 to 38.5 61.7 to 90.0 | 73.2 to 98.6 17.4 to 64.5 36.1 to 82.6 | 97.2 to 98.6 12.7 to 12.7 87.1 to 87.3 | 85.9 to 85.9 46.9 to 51.7 48.5 to 53.4 |
| No. of measures with <50% score 4 | 0 | 9 | 0 | 6 | 0 | 3 | 0 | 1 |

Note: a = Measure P7 (% active patients aged 0–19 years screened for adverse childhood experiences in previous 12 months) did not achieve agreement in Round 1.

(% active patients aged 0–19 years screened for adverse childhood experiences in previous 12 months) (achieved only 68.2% in feasibility after Round 1, but this was adjusted to 71.1% following re-rating in Round 3. Note that an active patient is defined as "a patient who has attended the practice/service three or more times in the past two years [22].

The overall mean relevance scores were between 3.7 and 3.8, with range between 3.0 and 4.0; and both median and mode scores were 4 across all means. The mean feasibility scores were between 3.3 and 3.6, with range between 2.9 and 4.0; and both median and mode scores were 3 and 4 across all means.

A skewness value greater than 1 or less than −1 indicates a highly skewed distribution. Negative values for skewness indicate data that are skewed left, i.e., the left tail in the x axis is long relative to the right tail [41]. Even though the skewness ranges of the mean scores for all measures indicate a strong left skewness, i.e., the distribution of scores was largely in the higher numbered options, data for feasibility is less skewed than that for relevance in each of the attribute. When the scores of 3 and 4 were combined, the agreements for relevance were between 85.9% and 100.0% but for feasibility were between 71.1% and 100.0%.

### Measures that have reached consensus but commonly rated as 'somewhat feasible' rather than 'feasible'

Nineteen measures in 13 indicators were more commonly rated as 'somewhat feasible' rather than 'feasible' (Table 5). Nine of these are under Attribute 1 (accountability to our patients), seven under Attribute 2 (professionally accountable), three under Attribute 3 (accountability to the community), and none under Attribute 4 (accountability to society). The measure that had the lowest proportion that rated 'feasible' was Measure O12e, followed by Measure O57b and Measure P7a. Eight of the 19 measures are the original "blue sky" measures, and ten are related to patient-reported measures.

**Sub-group analyses.**

### Comparison of ratings between practice staff and PHN staff

Table 6 shows the comparisons of the overall rating by practice and PHN staff across all four attributes.

Distribution of the mean scores for relevance and feasibility for both groups were similar, although some of the feasibility scores by PHN staff in attributes 1 and 2, when compared with those of practice staff, were less skewed towards the higher numbered options.

When the scores of 3 and 4 were combined, the mean scores by PHN staff were generally higher than those by practice staff. At Round 3, although practice staff pulled their rating of P7 to 71.4%, PHN staff rating of P7 only pulled up to only 69.2%, slightly shy of the 70% threshold.

And when we calculated the proportions of ratings 3 and 4, 21 measures were more commonly rated by practice staff as 'somewhat feasible' rather than 'feasible' compared with 26 measures by PHN staff. Thirteen of these were the same and seven of those 13 were the original "blue sky" measures. The measure that had the lowest proportion of 'feasible' rating by practice staff was Measure O12e; and the lowest by PHN staff was Measure S73a. Only one measure (Indicator S65: Registered for postgraduate GP training – Measure S65a: Accredited as training practice with local RTO) was more commonly rated by PHN staff as 'somewhat relevant' rather than 'relevant'.

### Comparison of ratings between clinician and non-clinician practice staff

Table 7 shows the comparisons of the overall rating by clinicians and non-clinicians in the practice staff across all four attributes.

Distribution of the mean scores for relevance and feasibility for both groups were similar, although the feasibility scores by both clinician and non-clinician practice staff were generally less skewed than their relevance scores towards the higher numbered options.

When the scores of 3 and 4 were combined, the mean scores by non-clinician staff were generally higher than those by clinician staff. At Round 3, although non-clinician staff adjusted their rating of P7 to 71.4%, clinician staff rating of P7

**Table 5. Measures that have reached consensus in feasibility but more often rated as 'somewhat feasible' rather than 'feasible'.**

| | | Combined 3 and 4 scores (%) | Proportion 3 scores 'somewhat feasible' (%) | Proportion 4 scores 'feasible' (%) |
|---|---|---|---|---|
| **ATTRIBUTE 1** | | | | |
| Indicator O4: Patient activation – Measure O4a: PAM®[1] scores (BLUE SKY) | | 80.8 | 60.5 | 39.5 |
| Indicator P6: Risk factors recorded – Measure P6e: % active patients aged 14–19 years with other substance[2] use recorded | | 76.1 | 53.8 | 46.3 |
| Indicator P7: Childhood adverse experiences recorded – Measure P7a: % active patients aged 0–19 years screened for adverse childhood experiences in previous 12 months (BLUE SKY) | | 71.1 | 65.7 | 34.3 |
| Indicator O12: Patient perceptions of preventative health discussion – PREMs[3] | Measure O12a: Healthy eating | 87.4 | 61.8 | 38.1 |
| | Measure O12b: Exercise/physical activity | 85.2 | 58.7 | 41.3 |
| | Measure O12e: Unintentional injuries (home risk factors) | 80.7 | 66.2 | 33.8 |
| | Measure O12f: Unsafe sexual practices | 77.3 | 60.3 | 39.7 |
| | Measure O12g: Unmanaged psychosocial stress | 81.8 | 61.1 | 38.9 |
| Indicator S43: Advance care planning – Measure S43a: % active patients ≥75 years with discussions about advance care planning recorded on file (BLUE SKY) | | 86.3 | 57.9 | 42.0 |
| **ATTRIBUTE 2** | | | | |
| Indicator P53: Team-based care – Measure P53a: Regular clinical review meetings involving all team members | | 93.1 | 51.5 | 48.5 |
| Indicator O55: GP and staff satisfaction – survey measuring GP and staff satisfaction with – Measure O55f: Work/life balance | | 91.8 | 53.7 | 46.3 |
| Indicator O57: Care plan engages patient | Measure O57a: PREM[3] questions on patients reporting of experience with care planning | 87.7 | 53.1 | 46.9 |
| | Measure O57b: PAM®[1] scores (BLUE SKY) | 80.8 | 66.1 | 33.9 |
| Indicator O58: Follow-up following hospital attendance | Measure O58a: % of active patients reviewed following emergency department presentation within 7 days (BLUE SKY) | 89.0 | 55.4 | 44.6 |
| | Measure O58b: % of active patients reviewed following admission within 3 days (BLUE SKY) | 83.6 | 65.6 | 34.4 |
| Indicator O64: Consumer satisfaction with quality of care – Measure O64a: Analysis of validated survey responses (PREMs[3]) (BLUE SKY) | | 93.1 | 58.2 | 41.8 |
| **ATTRIBUTE 3** | | | | |
| Indicator S73: Community engagement – Measure S73a: Practice has community/patient advisory structures | | 73.6 | 64.2 | 35.8 |
| Indicator O76: Access to regular primary care provider (as measured in response to PREMs[3]) – Measure O76b: % active patients reporting difficulties obtaining care in previous 12 months | | 86.1 | 51.6 | 48.4 |
| Indicator O77: Access for low socioeconomic status – Measure O77a: Compare % active patients who are Australian Government health care card holders with % holding Australian Government health care cards in practice LGA[4] (BLUE SKY) | | 88.9 | 57.8 | 42.2 |

Notes: 1 = Patient Activation Measure (PAM®) is a 10 or 13 item survey used to assess an individual's knowledge, skills and confidence in managing their own health and healthcare [42]; 2 = substances other than cigarettes and alcohol; 3 = Patient-reported experience measures (PREMs) collect information about patient's experiences of health services [43]; 3 = Local government area (LGA) is an Australian Bureau of Statistics approximation of gazetted local government boundaries as defined by each state and territory [44].

was only 66.6%. There were other measures in attributes 2 and 3 that clinicians rated below 70% such as Indicator S73: Community engagement – Measure S73a: Practice has community/patient advisory structures.

Twenty-six measures were more commonly rated by practice clinician staff as 'somewhat feasible' rather than 'feasible' compared with 20 measures by non-clinician staff; 10 of which are the same and four of these are the original "blue sky"

**Table 6. Comparison between practice and PHN staff.**

| | Attribute 1 (47 indicators with 79 measures) | | | | Attribute 2 (19 indicators with 31 measures) | | | | Attribute 3 (10 indicators with 16 measures) | | | | Attribute 4 (2 indicators with 2 measures) | | | |
|---|---|---|---|---|---|---|---|---|---|---|---|---|---|---|---|---|
| | Relevance | | Feasibility | | Relevance | | Feasibility | | Relevance | | Feasibility | | Relevance | | Feasibility | |
| | Practice | PHN | Practice | PHN | Practice | PHN | Practice | PHN | Practice | PHN | Practice | PHN | Practice | PHN | Practice | PHN |
| No. of raters | 54-68 | 26 | 54-68 | 26 | 45-46 | 26 | 45-46 | 26 | 45 | 26 | 45 | 26 | 45 | 26 | 45 | 26 |
| No. of measures reached consensus after 1st rating | 79 of 79 | 79 of 79 | 78 of 79 | 78a of 79 | 31 of 31 | 31 of 31 | 31 of 31 | 31 of 31 | 16 of 16 | 16 of 16 | 16 of 16 | 16 of 16 | 2 of 2 | 2 of 2 | 2 of 2 | 2 of 2 |
| Proportion consensus of P7 after 2nd rating | N/A | N/A | 1 of 1 | 0 of 1 | N/A | N/A | N/A | N/A | N/A | N/A | N/A | N/A | N/A | N/A | N/A | N/A |
| Mean score ± standard deviation | 3.8±0.1 | 3.9±0.1 | 3.5±0.2 | 3.6±0.2 | 3.8±0.1 | 3.8±0.1 | 3.5±0.2 | 3.5±0.3 | 3.7±0.1 | 3.8±0.2 | 3.5±0.2 | 3.5±0.3 | 3.8±0.0 | 3.9±0.0 | 3.3±0.0 | 3.3±0.1 |
| Mean score range | 3.4-4.0 | 3.5-4.0 | 2.9-3.8 | 2.8-3.9 | 3.3-4.0 | 3.3-4.0 | 3.1-3.9 | 3.0-4.0 | 3.5-3.9 | 3.4-4.0 | 3.0-3.7 | 2.8-3.9 | 3.7-3.8 | 3.9-4.0 | 3.2-3.3 | 3.2-3.4 |
| Median scores | 4 | 4 | 3 and 4 | 3 and 4 | 4 | 4 | 3 and 4 | 3 and 4 | 4 | 4 | 3 and 4 | 3 and 4 | 4 | 4 | 3 | 3.0 and 3.5 |
| Mode scores | 4 | 4 | 3 and 4 | 3 and 4 | 4 | 4 | 3 and 4 | 3 and 4 | 4 | 4 | 3 and 4 | 3 and 4 | 4 | 4 | 4 | 3.0 and 3.5 |
| Skewness range | −5.2 to −1.0 | −5.1 to −0.9 | −1.9 to −0.4 | −4.0 to 0.2 | −6.8 to −0.9 | −5.1 to −1.1 | −2.9 to−0.1 | −5.1 to 0.5 | −3.2 to −0.9 | −5.1 to 0.0 | −2.3 to −0.3 | −2.6 to −0.2 | −3.9 to −2.0 | −5.1 to −2.6 | −1.1 to −0.8 | −0.7 to −0.3 |
| Score 1 (% range) | 0.0-2.9 | 0.0-3.8 | 0.0-5.6 | 0.0-3.8 | 0.0-11.1 | 0.0-3.8 | 0.0-8.9 | 0.0-3.8 | 0.0-2.2 | 0.0 | 0.0-4.4 | 0.0-7.7 | 0.0-2.2 | 0.0-0.0 | 2.2-4.4 | 0.0 |
| Score 2 (% range) | 0.0-8.8 | 0.0-7.7 | 0.0-23.8 | 0.0-30.8 | 0.0-8.7 | 0.0-7.7 | 0.0-26.1 | 0.0-19.2 | 0.0-13.3 | 0.0-11.5 | 2.2-26.7 | 0.0-19.2 | 0.0-4.4 | 0.0-0.0 | 8.9-13.3 | 11.5-15.4 |
| Score 3 (% range) | 3.2-33.8 | 0.0-34.6 | 19.3-54.8 | 3.8-57.7 | 2.2-37.8 | 0.0-50.0 | 11.1-54.3 | 0.0-69.2 | 8.9-33.3 | 3.8-50.0 | 17.8-51.1 | 11.5-65.4 | 11.1-17.8 | 3.8-11.5 | 40.0-42.2 | 38.5-50.0 |
| Score 4 (% range) | 54.4-69.5 | 57.7-100.0 | 25.8-78.9 | 11.5-92.3 | 55.6-97.8 | 42.3-100.0 | 30.4-88.9 | 23.1-100.0 | 57.8-91.1 | 50.0-96.2 | 35.6-75.6 | 11.5-88.5 | 77.8-86.7 | 88.5-96.2 | 42.67-46.7 | 34.6-50.0 |
| Combined scores 3 and 4 (% range) | 88.2-100.0 | 92.3-100.0 | 71.4-100.0 | 69.2-100.0 | 84.4-100.0 | 88.5-100.0 | 73.9-100.0 | 80.8-100.0 | 84.4-100.0 | 88.5-100.0 | 71.1-97.8 | 76.9-100.0 | 95.6-97.8 | 100.0-100.0 | 84.4-86.7 | 84.6-88.5 |
| Proportion that is rating 3 (% range) | 3.3-38.3 | 0.0-37.5 | 20.0-68.0 | 4.0-83.4 | 2.2-40.5 | 0.0-54.2 | 11.1-64.1 | 0.0-75.0 | 8.9-36.6 | 3.8-50.0 | 19.0-56.1 | 11.5-85.0 | 11.4-18.6 | 3.8-11.5 | 46.2-50.0 | 43.5-59.1 |
| Proportion that is rating 4 (% range) | 61.7-96.7 | 62.5-100.0 | 32.0-80.0 | 16.7-96.0 | 59.5-97.8 | 45.8-100.0 | 35.9-88.9 | 25.0-100.0 | 63.4-91.1 | 50.0-96.2 | 43.9-81.0 | 15.0-88.5 | 81.4-88.6 | 88.5-96.2 | 50.0-53.8 | 40.9-56.5 |
| No. of measures with <50% score 4 | 0 | 0 | 24 | 19 | 0 | 1 | 11 | 12 | 0 | 0 | 5 | 6 | 0 | 0 | 2 | 1 |

Note: a = Measure P7 (% active patients aged 0–19 years screened for adverse childhood experiences in previous 12 months) did not achieve agreement in Round 1.

**Table 7. Comparison between clinicians and non-clinicians in the practice staff.**

| | Attribute 1 (47 indicators with 79 measures) | | | | Attribute 2 (19 indicators with 31 measures) | | | | Attribute 3 (10 indicators with 16 measures) | | | | Attribute 4 (2 indicators with 2 measures) | | | |
|---|---|---|---|---|---|---|---|---|---|---|---|---|---|---|---|---|
| | Relevance | | Feasibility | | Relevance | | Feasibility | | Relevance | | Feasibility | | Relevance | | Feasibility | |
| | Clin | Non-clin | Clin | Non-clin | Clin | Non-clin | Clin | Non-clin | Clin | Non-clin | Clin | Non-clin | Clin | Non-clin | Clin | Non-clin |
| No. of raters | 27-33 | 25-32 | 27-33 | 25-32 | 22-23 | 22 | 22-23 | 22 | 22 | 22 | 22 | 22 | 22 | 22 | 22 | 22 |
| No. of measures reached consensus after 1st rating | 79 of 79 | 79 of 79 | 78ᵃ of 79 | 79 of 79 | 31 of 31 | 31 of 31 | 26 of 31 | 31 of 31 | 16 of 16 | 16 of 16 | 15 of 16 | 16 of 16 | 2 of 2 | 2 of 2 | 2 of 2 | 2 of 2 |
| No. of measures reached consensus after 2nd rating | N/A | N/A | 1 of 1 | 1 of 1 | N/A | N/A | N/A | N/A | N/A | N/A | N/A | N/A | N/A | N/A | N/A | N/A |
| Mean score±standard deviation | 3.8±0.1 | 3.8±0.1 | 3.5±0.2 | 3.5±0.2 | 3.7±0.2 | 3.8±0.2 | 3.4±0.2 | 3.6±0.2 | 3.7±0.1 | 3.8±0.2 | 3.4±0.2 | 3.5±0.2 | 3.8±0.1 | 3.8±0.0 | 3.2±0.1 | 3.4±0.1 |
| Mean score range | 3.4-4.0 | 3.3-4.0 | 2.8-3.8 | 3.0-3.9 | 3.3-4.0 | 3.4-4.0 | 2.9-3.8 | 3.1-4.0 | 3.4-3.9 | 3.4-4.0 | 3.0-3.7 | 3.1-3.9 | 3.7-3.9 | 3.7-3.8 | 3.1-3.3 | 3.3-3.5 |
| Median scores | 4 | 3 and 4 | 3 and 4 | 3 and 4 | 3 and 4 | 4 | 3 and 4 | 3 and 4 | 4 | 3.5 and 4 | 3 and 4 | 3 and 4 | 4 | 4 | 3 and 3.5 | 3 and 3.5 |
| Mode scores | 4 | 4 | 3 and 4 | 3 and 4 | 3 and 4 | 3 and 4 | 3 and 4 | 3 and 4 | 4 | 3 and 4 | 3 and 4 | 3 and 4 | 4 | 4 | 3 and 4 | 3 and 4 |
| Skewness range | −5.4 to −0.6 | −5.2 to −0.6 | −2.1 to −0.1 | −2.7 to −0.1 | −4.8 to −0.9 | −4.7 to −0.8 | −2.5 to 0.2 | −4.7 to 1.1 | −3.1 to −1.1 | −4.7 to −0.7 | −2.0 to −0.3 | −2.3 to −0.1 | −2.3 to −2.0 | −3.6 to −2.0 | −1.1 to −0.7 | −1.3 to −0.6 |
| Score 1 (% range) | 0.0-3.7 | 0.0-3.7 | 0.0-11.1 | 0.0-8.0 | 0.0-8.7 | 0.0-13.6 | 0.0-8.7 | 0.0-9.1 | 0.0-4.3 | 0.0-4.5 | 0.0-4.3 | 0.0-4.5 | 0.0 | 0.0-4.5 | 4.3 | 0.0-4.5 |
| Score 2 (% range) | 0.0-12.1 | 0.0-14.8 | 0.0-28.6 | 0.0-25.0 | 0.0-13.0 | 0.0-9.1 | 0.0-34.8 | 0.0-13.6 | 0.0-13.0 | 0.0-13.6 | 0.0-30.4 | 0.0-22.7 | 0.0 | 0.0-4.5 | 13.0-17.4 | 4.5 |
| Score 3 (% range) | 0.0-37.0 | 0.0-43.8 | 17.2-56.3 | 11.1-62.5 | 4.3-43.5 | 0.0-36.4 | 13.0-56.5 | 4.5-72.7 | 8.7-30.4 | 4.5-36.4 | 17.4-52.2 | 13.6-59.1 | 13.0-17.4 | 9.1-18.2 | 30.4-39.1 | 45.5 |
| Score 4 (% range) | 57.6-100.0 | 46.9-100.0 | 18.8-82.8 | 28.1-88.0 | 43.5-95.7 | 59.1-100.0 | 26.1-82.6 | 22.7-95.5 | 60.9-87.0 | 50.0-95.5 | 34.8-69.6 | 27.3-86.4 | 73.9-82.6 | 77.3-86.4 | 34.8-47.8 | 45.5-50.0 |
| Combined scores 3 and 4 (% range) | 84.9-100.1 | 85.1-100.0 | 66.6-100.1 | 71.4-100.0 | 78.2-100.0 | 86.3-100.0 | 65.2-100.0 | 81.8-100.0 | 78.3-95.7 | 86.3-100.0 | 60.8-95.7 | 77.3-100.0 | 91.3-95.6 | 95.5 | 73.9-78.2 | 91.0-95.5 |
| Proportion that is rating 3 (% range) | 0.0-37.0 | 0.0-48.3 | 17.2-72.7 | 11.5-69.0 | 4.3-50.0 | 0.0-38.1 | 13.6-65.0 | 4.5-75.0 | 9.1-33.3 | 4.5-42.1 | 20.0-60.0 | 13.6-68.4 | 13.6-19.1 | 9.5-19.1 | 38.9-52.9 | 47.6-50.0 |
| Proportion that is rating 4 (% range) | 63.0-100.0 | 51.7-100.0 | 27.3-82.8 | 31.0-88.5 | 50.0-95.7 | 61.9-100.0 | 35.0-86.4 | 25.0-95.5 | 66.7-90.9 | 57.9-95.5 | 40.0-80.0 | 31.6-86.4 | 80.9-86.4 | 80.9-90.5 | 47.1-61.1 | 50.0-52.4 |
| No. of measures with <50% score 4 | 0 | 2 | 22 | 18 | 1 | 0 | 20 | 8 | 0 | 0 | 8 | 3 | 0 | 0 | 2 | 1 |

Note: a = Measure P7 (% active patients aged 0–19 years screened for adverse childhood experiences in previous 12 months) did not achieve agreement in Round 1.

measures. The measure that had the lowest proportion of 'feasible' rating by clinician staff was Measure O12f; and the lowest by non-clinician staff was Measure 057b.

**Measures more commonly rated as 'somewhat feasible' than 'feasible'.** Table 8 shows all measures more commonly rated as 'somewhat feasible' than 'feasible' both overall and by the subgroups. Eight of these are common to all subgroups, six of which are related to patient reported measures and only three of these were the original "blue sky" measures designated during the indicators' development stage.

## Comparison of agreement between the subgroups

Round 1 presented, initially, 4371 pairs of scores, while round 2 had 2556 pairs. After the exclusion of raters who scored less than 90% of items, round 1 had 3160 valid comparisons (72.3%); and Round 2 had 2485 valid comparisons (97.2%).

Comparing practice staff with PHN staff and clinician practice staff with non-clinician practice staff, all agreements were considered statistically significant as can be observed in the p values and the small confidence intervals, although feasibility agreement between clinicians and non-clinicians are slightly outside the threshold of $p = 0.05$. (Table 9) In general, the mean agreements indicate moderate agreements in relevance and low agreements in feasibility irrespective of the subgroups. Also, relevance shows an agreement around 50% more than feasibility. Although statistically significant, the observed differences between sub-groups are small from the practice perspective and are, possibly, an artefact created by the large sample sizes in this study.

## Qualitative results

Four main themes were elicited with multiple subthemes (Table 10). Detailed data and selected quotes are available in S3 File.

**Theme 1: Use of QUEST PHC indicators and measures.** Survey respondents generally recognised the importance of quality indicators. They acknowledged that the QUEST PHC measures would potentially provide opportunities to reflect the high-quality care already offered in general practice, and help benchmark care standards and identify gaps in services to further drive quality particularly through quality improvement activities and support provided by PHNs.

> "I think the indicators and measures reflect high-quality care quite well. I think for a lot of practices these goals would be suitably aspirational and achievable."

> "It will assist GP practice to identify services gap if there's any and then provide better quality of care accordingly, patient satisfaction will be higher too."

> "It would be a useful tool for practice managers, GPs, nurses enabling and assisting with the work and support offered by PHNs."

Participants considered the tool would help general practice to self-reflect on care provision as well as seek feedback from patients and consumers.

> "Data should be made available to all staff who could be guided in reflection on them to help effect positive change."

> "Patient activation and PREM (patient-reported experience measures) aren't currently being used in general practice broadly. Would be great to see these items used in a way that's meaningful to patients and to practices."

Although incentive-driven funding and resource allocation in primary care were seen to be potential benefits of the tool, more importantly, it was seen to facilitate improvement in patient-centred care, patient engagement and empowerment, and patient health outcomes.

**Table 8. Measures that have reached consensus in feasibility but rated > half the time as 'somewhat feasible' compared to 'feasible'.**

| Indicator | All participants | PHN staff | Practice staff | Clinicians | Non-clinicians |
|---|---|---|---|---|---|
| *Indicator O4: Patient activation – Measure O4a: Patient Activation Measure® scores (BLUE SKY)* | *60.5* | *52.6* | *63.2* | *53.8* | *69.0* |
| Indicator P6: Risk factors recorded – Measure P6a: % active patients ≥15 years with a BMI recorded who have weight classification (obese, overweight, healthy, underweight) in previous 12 months | | | 52.5 | | 57.7 |
| Indicator P6: Risk factors recorded – Measure P6b: % active patients ≤15 years with height/length and weight recorded in previous 12 months | | | | 52.0 | |
| Indicator P6: Risk factors recorded – Measure P6e: % active patients aged 14–19 years with other substance use recorded | 53.8 | | 56.3 | 54.1 | 57.1 |
| *Indicator P7: Childhood adverse experiences recorded – Measure P7a: % active patients aged 0–19 years screened for adverse childhood experiences in previous 12 months (BLUE SKY)* | *65.7* | *83.4* | *60.0* | *62.5* | *55.0* |
| Indicator P11: Aboriginal and Torres Strait Islander preventative health care Measure P11a: % active patients identified as Aboriginal or Torres Strait Islander with Aboriginal health check in previous 15 months | | | | 55.6 | |
| *Indicator O12: Patient perceptions of preventative health discussion – Patient Reported Experience – Measure O12a: Healthy eating* | *61.8* | *60.0* | *62.7* | *66.7* | *60.9* |
| *Indicator O12: Patient perceptions of preventative health discussion – Patient Reported Experience – Measure O12b: Exercise/physical activity* | *58.7* | *54.2* | *60.8* | *65.4* | *56.5* |
| Indicator O12: Patient perceptions of preventative health discussion – Patient Reported Experience – Feasibility – Measure O12c: Risks of smoking/quit smoking | | | 54.6 | 64.3 | |
| Indicator O12: Patient perceptions of preventative health discussion – Patient Reported Experience – Measure O12d: Alcohol use | | | 55.6 | 62.9 | |
| *Indicator O12: Patient perceptions of preventative health discussion – Patient Reported Experience – Measure O12e: Unintentional injuries (home risk factors)* | *66.2* | *61.9* | *68.0* | *68.0* | *65.3* |
| *Indicator O12: Patient perceptions of preventative health discussion – Patient Reported Experience – Measure O12f: Unsafe sexual practices* | *60.3* | *52.2* | *64.4* | *72.7* | *54.5* |
| *Indicator O12: Patient perceptions of preventative health discussion – Patient Reported Experience – Measure O12g: Unmanaged psychosocial stress* | *61.1* | *52.2* | *65.3* | *72.0* | *56.5* |
| Indicator P19: Diabetes – managing complications – Measure P19a: % active patients with DM and have their retinal screening performed in previous 24 months | | | | | 54.2 |
| Indicator P24: Respiratory disease – use of spirometry record – Measure P24b: % active patients with asthma and have their spirometry recorded in previous 24 months | | | 53.3 | 57.2 | |
| Indicator P27: Respiratory disease – control – Measure P27a: % active patients with COPD and have their COPD Assessment Test score *(BLUE SKY)* | | | | 52.2 | |
| Indicator P27: Respiratory disease – control – Measure P27b: % active patients with asthma and have Asthma Control Questionnaire recorded *(BLUE SKY)* | | | | 52.2 | |
| Indicator S43: Advance care planning – Measure S43a: % active patients ≥75 years with discussions about advance care planning recorded on file *(BLUE SKY)* | 57.9 | 56.5 | 58.7 | 60.9 | |
| Indicator S48: Practice goal/mission – Measure S48a: Defined practice mission/goal | | | | 57.2 | |
| Indicator S48: Practice goal/mission – Measure S48c: Mission/goal accessible to patients | | | | 65.0 | |
| Indicator S50: Data sharing with local hospitals – Measure S50b: Able to receive data in real time, e.g., shared EHR or real time electronic shared care plan *(BLUE SKY)* | | 65.2 | | | |
| Indicator S51: Data sharing with other health care providers – Measure S51a: Practice has a system for notifying GPs of specialist and allied health care correspondence *(BLUE SKY)* | | 60.0 | | | |
| Indicator P53: Team-based care – Measure P53a: Regular clinical review meetings involving all team members | 51.5 | 61.5 | | | |

*(Continued)*

**Table 8.** (Continued)

| Indicator | All participants | PHN staff | Practice staff | Clinicians | Non-clinicians |
|---|---|---|---|---|---|
| Indicator P53: Team-based care – Measure P53b: Assigned care teams to coordinate care for individual patients *(BLUE SKY)* | | | 56.0 | 60.0 | 52.7 |
| Indicator O55: GP and staff satisfaction – survey measuring GP and staff satisfaction with: – Measure O55b: Impact on local community | | 59.1 | | | |
| Indicator O55: GP and staff satisfaction – survey measuring GP and staff satisfaction with: – Measure O55d: Income from work | | | | 52.4 | |
| Indicator O55: GP and staff satisfaction – survey measuring GP and staff satisfaction with: – Measure O55f: Work/life balance | 53.7 | 52.2 | 53.7 | | 61.9 |
| Indicator O57: Care plan engages patient – Measure O57a: Patient Reported Experience Measure (PREM) questions on patients reporting of experience with care planning | 53.1 | | 56.4 | | 72.7 |
| Indicator O57: Care plan engages patient – Measure O57b: Patient activation measure (PAM®) scores *(BLUE SKY)* | 66.1 | 72.7 | 61.1 | | 75.0 |
| Indicator O58: Follow-up following hospital attendance – Measure O58a: % of active patients reviewed following ED presentation within 7 days *(BLUE SKY)* | 55.4 | 52.2 | 56.1 | | 66.7 |
| *Indicator O58: Follow-up following hospital attendance – Measure O58b: % of active patients reviewed following admission within 3 days (BLUE SKY)* | *65.6* | *66.7* | *64.1* | *61.1* | *66.7* |
| Indicator P61: Assessment of learning needs – Measure P61a: Evidence of process for assessment of learning needs | | 52.0 | | | 52.4 |
| Indicator O64: Consumer satisfaction with quality of care – Measure O64a: Analysis of validated survey responses (PREMs) *(BLUE SKY)* | 58.2 | 75.0 | | 57.9 | |
| Indicator S65: Registered for postgraduate GP training – Measure S65a: Accredited as training practice with local RTO | | 73.9 | | | |
| Indicator P66: Engagement with student training – Measure P66a: Number of medical, nursing and allied health students undertaking placements in previous 12 months | | 63.6 | | | |
| Indicator P67: Research activity – Measure P67a: Evidence of engagement with research or Plan – Do – Study – Act activities *(BLUE SKY)* | | | 54.1 | 56.3 | |
| Indicator S68: Urgent access to care – Measure S68a: Provides same day appointments | | 61.0 | | | |
| Indicator S72: Health related social needs assessed – Measure S72a: % active patients with screening for health-related social needs recorded *(BLUE SKY)* | | | | | 52.4 |
| Indicator S73: Community engagement – Measure S73a: Practice has community/patient advisory structures | 64.2 | 85.0 | | | 64.7 |
| Indicator P75: Meets the needs of CALD communities – Measure P75a: Provides bilingual services as required | | 52.2 | | 60.0 | |
| Indicator O76: Access to regular primary care provider (as measured in response to Patient Reported Experience – Measure O76b: % active patients reporting difficulties obtaining care in previous 12 months | 51.6 | 54.2 | | 57.9 | |
| Indicator O77: Access for low socioeconomic status – Measure O77a: Compare % active patients who are Australian Government health care card holders with % holding Australian Government health care cards in practice LGA *(BLUE SKY)* | 57.8 | 59.1 | 56.1 | | 68.4 |
| Indicator O78: Avoidable hospital care – Measure O78a: Use of linked data to measure potentially preventable hospital admissions *(BLUE SKY)* | | | | 52.9 | |
| Indicator O79: Duplication of care – Measure O79a: Use of linked data to avoid duplication of pathology and radiology services *(BLUE SKY)* | | 59.1 | | | |

Note: Measures common to all subgroups are bolded and italicised.

**Table 9. Agreement between subgroups.**

| | Round 1 | | | | | Round 2 | | | | |
|---|---|---|---|---|---|---|---|---|---|---|
| | General | PHN | Practice | Clinician | Non-clinician | General | PHN | Practice | Clinician | Non-clinician |
| **RELEVANCE AGREEMENT** | | | | | | | | | | |
| **Mean agreement** | 0.76 | 0.73 | 0.85 | 0.78 | 0.66 | 0.70 | 0.67 | 0.75 | 0.63 | 0.72 |
| **LCI** | 0.75 | 0.72 | 0.83 | 0.76 | 0.63 | 0.69 | 0.66 | 0.73 | 0.59 | 0.70 |
| **UCI** | 0.77 | 0.74 | 0.87 | 0.80 | 0.68 | 0.71 | 0.69 | 0.77 | 0.66 | 0.74 |
| **Comparison** | NA | PHN x Practice | | Clinician x Non-clinician | | NA | PHN x Practice | | Clinician x Non-clinician | |
| **p-value** | NA | <0.001 | | <0.001 | | NA | <0.001 | | <0.001 | |
| **t-statistic** | NA | −12.255 | | 7.458 | | NA | −6.082 | | −4.906 | |
| **FEASIBILITY AGREEMENT** | | | | | | | | | | |
| **Mean agreement** | 0.48 | 0.47 | 0.53 | 0.48 | 0.44 | 0.48 | 0.46 | 0.52 | 0.42 | 0.52 |
| **LCI** | 0.47 | 0.46 | 0.51 | 0.46 | 0.41 | 0.47 | 0.45 | 0.51 | 0.40 | 0.50 |
| **UCI** | 0.49 | 0.48 | 0.55 | 0.49 | 0.47 | 0.49 | 0.47 | 0.53 | 0.45 | 0.53 |
| **Comparison** | NA | PHN x Practice | | Clinician x Non-clinician | | NA | PHN x Practice | | Clinician x Non-clinician | |
| **p-value** | NA | <0.001 | | 0.055 | | NA | <0.001 | | <0.001 | |
| **t-statistic** | NA | −5.464 | | 1.925 | | NA | −6.505 | | −6.256 | |

Note: LCI = lower 95% confidence interval; UCI = upper 95% confidence interval; NA = not applicable.

"Support the case for increased remuneration. Support investment in (their) practice workforce and infrastructure."

"General practice needs to evolve at a faster pace in implementing models of care that are truly patient centred resulting in improved patient experiences and outcomes."

There were, however, potential issues and concerns expressed about the feasibility of some measures, e.g., indicator 058 'follow- up following hospital attendance', which were reliant on timely and effective discharge communication from hospitals and other sources.

"Indicator O58 is only possible if general practices are able to receive timely and informative discharge summaries from hospital."

There were also suggestions to include other measures such as domestic violence and wound healing.

"An additional indicator for consideration is family violence."

"Determining GP competency in relation to undertaking various types of skin procedures. Wound healing outcomes especially concerning chronic wounds."

However, many respondents commented that some aspects of quality patient care, such as relationship and trust, could not be measured.

"I believe the hallmark of excellence in general practice is to provide a safe and non-judgemental space with a health-care provider team that they know and trust and where their personalised health outcomes are paramount at each consultation. It is hard to know how you measure this type of relationship with a tool."

**Table 10. Themes from qualitative responses.**

| Themes | Subthemes |
|---|---|
| **Use of QUEST PHC indicators and measures** | Reflect high-quality care |
| | Drive and support quality |
| | Support QI |
| | Seek feedback |
| | Benchmark |
| | Identify gaps |
| | Allocate resources in primary care |
| | Help reflection |
| | Help PHNs in their work with/support for practices |
| | Potential patient care outcomes of the tool |
| | Potential issues and concerns |
| **Barriers to using quality indicators and measures** | Systemic barriers |
| | GP factors |
| | Patient factors |
| | The reality of general practice |
| | Doubts about the tool - 'just ticking boxes' |
| | Benefits of tool unclear |
| | Concerns about how data will be used |
| | Concerns about PRMs |
| | Risks of alienating/demoralising practices |
| **Perceptions of equity of the QUEST PHC tool** | Equity of access across different populations |
| | Equity of the health system across different geographical areas |
| **Suggestions on implementation of the QUEST PHC tool** | Promotion and communication |
| | Must have buy-in |
| | Carrot, not stick |
| | Staff to be involved |
| | Integrate into current normal workflow or improve |
| | Implement an appropriate number of indicators |
| | Need a benchmarking/ standardisation component |
| | Need accurate datasets |
| | Data linkages needed |
| | Need to overhaul the healthcare system |
| | Need to be supported by PHNs |
| | Suggestions of features for the tool |
| | Evaluation of the tool |
| | Make it a national tool |

**Theme 2: Barriers to using quality indicators and measures.** Respondents described many systemic barriers to implementing what they perceived to be 'more' measurements in general practice, e.g., deficiencies of the current health system, lack of staff, time and funding, and issues with data linkages.

"MBS significantly undervalues large portions of the health sector and will not deliver on provider and patient satisfaction, it does not understand what is needed and how measure and appropriately remunerate complex, chronic care."

"Some providers would be hesitant to utilise it due to time and workforce constraints."

"…feasibility will be tied to inconsistent use of data systems and linkages between primary and tertiary health. e.g., inconsistent use of MHR in tertiary system."

There were also factors specifically related to GPs (e.g., perceived threat such as reduced government funding and lack of value or benefits of the tool) and patients (e.g., lack of understanding of the value of the measures and their feedback to help improve their care).

"GPs might perceive it as a threat and resist. Government may use such a tool as a funding lever. General practice not given the skills to undertake quality improvement."

"Patients have to have an understanding of WHY this is important for their healthcare otherwise low uptake might be the result."

The reality of general practice being small businesses was mentioned several times.

"…it's important for measures to consider that practices are small businesses & it's not uncommon for practices to want to meet measures but where the bottom dollar is a barrier to time & effort."

There were concerns that the tool could be just another 'box-ticking' exercise, about the lack of clarity regarding how data collected would be used, and how patient-reported measures (PRMs) would be implemented.

"I still have reservations about who is collecting demographic and SES data and how that data is being wielded to substantiate funding models and galvanizing blame games."

"It (PRMs) would be useful as a screening tool, but there are a few things to consider: Support the patient's right to refuse to participate. Consider the setting in which the data is collected, to ensure comfort and privacy. There are time constraints to factor in."

**Theme 3: Perceptions regarding suitability of the QUEST PHC tool for different populations.** There were questions amongst some respondents about whether the tool would be equitably accessible across different populations (including, adolescents, elderly people, people with low health literacy, culturally and linguistically diverse (CALD) patients and other vulnerable groups.) and different geographical areas (including remote, rural and metropolitan).

"Failure to capture the inherent difficulty of different patient populations. Focusing on absolute outcomes rather than relative improvement."

"Also does not take geopolitical aspects in delivering care city versus rural, local versus federal government constraints."

**These 4: Suggestions about Implementation of the QUEST PHC tool.** Survey respondents had many suggestions about how the tool should be implemented. They emphasised the tool should be 'carrot' not 'stick' for general practice to comply, and the need for effective promotion and communication before implementation.

"The tool would be great but beware that it does not become the stick that beats the last ounce of real care out of general practice."

Qualitative measures are not looked at enough and the role of engagement and trust and transparency would help practices have that conversation about what quality care means. What it requires to deliver care and hopefully frame the conversation in a shared language, so everyone is speaking to the same things."

Practice staff should be trained adequately, and there must be effective 'champions' (such as nurses and practice managers).

"Need to ensure staff buy in and appropriate training to address outcomes from tool with the practice."

The tool must be easy, manageable and able to be integrated into current practice workflow. Data linkage and accurate datasets would be critical to the effectiveness of the tool.

"Need to link it with morbidity and mortality data and use of health services."

"A tool that is user friendly and that could be incorporated into everyday work process than it could be fundamentally used by most practice staff."

It should be benchmarked or standardised prior to introduction and its use should be evaluated regularly.

"Standardising measures and ensuring they can be applied to all general practices (from solo to large practices and for metro to rural or very remote). Some practices do find RACGP 5th Edition standards favour larger practices and making it universal for any practice no matter size or location."

Respondents also remarked that the tool should be implemented nationally with support from PHNs to overhaul the current Australian health system.

"…requires financing for training in QI, workflows then financing for support to implement improvements, then support for better integration of care in primary/secondary and tertiary sectors, then shared responsibility across the whole health sector for health improvement."

Enabling the data to be reported on a practice dashboard, and use of a centralised data repository, potentially a cloud system was also suggested.

"A dashboard highlighting the measures would be useful."

"Need a centralised data based/ possibility cloud system."

## Discussion

This Delphi study aimed to validate the content of the QUEST PHC suite of indicators and measures for the Australian context by establishing consensus with general practice and PHN staff. Following three rounds of survey, consensus was reached by participating general practice and PHN staff for all 79 indicators and 128 measures based on the criteria set in the study protocol.

Consensus was easily achieved for relevance, but feasibility assessment was more complicated in our research. All indicators and measures reached consensus for relevance after the first round; only one measure (P7: % active patients aged 0–19 years screened for adverse childhood experiences in previous 12 months) needed to be re-rated for feasibility. Although this measure eventually reached the 70% threshold for consensus, the feasibility rating by clinician practice staff and PHN staff were lower than that of non-clinician practice staff.

The rating of all measures was high for both relevance and feasibility; however, 19 measures were more often rated 'somewhat feasible' rather than 'feasible'. The QUEST PHC measures are likely to be relevant since they were developed based on extensive literature review and consultations. However, our qualitative findings suggest their feasibility depends on many external "real world" factors including constraints of the current health system, issues with existing primary and secondary health data linkage, and general practice buy-in, capacity and time. It is worth noting that most of the 26 original measures designated "blue sky" during the development of the QUEST PHC suite were not regarded as such by the majority of our raters in this Delphi research. This could be explained by much improved processes such as care planning, team-based care, utilisation of My Health Records, and linkages with disease registries making a positive impact on data collection over the last few years since the QUEST PHC indicators and measures were developed. Only three "blue sky" measures were rated more often as 'somewhat feasible' rather than 'feasible' across the subgroups of PHN staff, practice staff, clinician practice staff and non-clinician practice staff. These measures are Measure O4a: Patient activation measure (PAM®) scores (blue sky) to assess person-centred care and patient-team relationship, Measure P7a: % active patients aged 0-19 years screened for adverse childhood experiences in previous 12 months, and Measure O58b: % of active patients reviewed following hospital admission within 3 days. An explanation is that PAM® and adverse childhood experience may be new and unfamiliar concepts for Australian general practice. A literature review examined the enablers and barriers to implementation of PAM® found that the organisations, clinicians and patients' perceived value and function of the PAM® influenced implementation and use. [45] Challenges with data linkage have always been a barrier to general practices receiving timely and quality information about patients' discharge from hospitals hence the perceived infeasibility of conducting a prompt post-hospital review [46,47].

However, despite the higher-than-expected ratings for feasibility across the QUEST PHC indicators and measures, our participants did describe many systemic, GP-specific and patient-specific barriers in their qualitative responses. Extensive advocacy, consultations and co-design with general practice, increased capacity-building and support for primary health care professionals as well as considerations for the 'small business' model of Australian general practice would be necessary for implementation of a tool such as the QUEST PHC indicators and measures.

There appeared to be some uncertainty about the best use of PRMs in measuring quality in general practice. Amongst the 19 measures more often considered 'somewhat feasible' rather than 'feasible' (such as O57b - record of PAM scores to assess patient engagement with care plan; O12e - PREMs to include patient report of discussion regarding home risk factors), eight were related to PRMs even though only 17 measures in total across the entire QUEST PHC suite of measures were related to PRMs. Most of the eight PRM-related measures were about patient perceptions of preventive health discussions. Participants' qualitative responses reflected their concerns about the use of PRMs with patients who may not understand the value of such measures and the impact of their feedback. Such concerns tied in with barriers described relating to the lack of time and staff capacity to explain to patients about quality measures or request their input. Efforts to design a process for general practice and consumers that would normalise PRMs as part of data collection will require collaboration from practices, PHNs and consumer advocates.

There is a growing recognition in recent years of the urgent need for funding reform in Australian general practice. Stream 2 of the Australia's Primary Health Care 10-Year Plan 2022–2032 outlines the government's plans to achieve "person-centred primary health care, supported by funding reform" and proposes to leverage "the voluntary patient registration (VPR) as a platform for reforming funding to incentivise quality person-centred primary health care" [48]. However, the development of both the VPR model and the related funding reform need to be informed by evidence and supported by ongoing evaluation of outcomes. Furthermore, if the Australian Government continues to invest in PIP QI, which currently focuses mainly on structures and processes and addresses only minimum quality standards, then evidence-based, professionally endorsed expansion of the PIP QI is essential. The QUEST PHC suite of indicators and measures have reached consensus in this Delphi study and could provide a framework for defining and measuring high-quality general practice. This would enable reporting to inform quality improvement as well as provide a basis for funding reforms rewarding high quality general practice and supporting primary healthcare providers.

The agreements between practice and PHN staff and between non-clinician and clinician practice staff were modest but statistically significant. The very small differences were nevertheless unlikely to be important in the "real world. This provides confidence that our sample selection in this study was appropriate, and our results did not occur by chance.

This study had several strengths. The rigorous and systematic approach used, the subgroup analyses performed and the inclusion of qualitative responses in the survey all added layers of additional information that provide a comprehensive picture of the validity of the indicators and measures as well as informing their future implementation in the real world. The project's strong governance structure and partnership with the PHNs provides opportunity for further development and roll-out of the QUEST PHC indicators and measures in Australian general practice. The delta COVID-19 outbreaks and the natural disasters of floods occurred in several states in Australia during the time of the study, and the demands on general practice and PHNs in emergency responses impacted significantly on the study recruitment, retention and possibly on interpretations of responses. Additionally, despite support from participating PHNs with strong rural focus and the Australian College of Rural and Remote Medicine, there was minimal response from rural primary health care professionals in the study. Although issues concerning rural practices and populations were sometimes raised by PHN staff, future consultations with rural practitioners should be conducted.

## Conclusions

The QUEST PHC suite of 79 indicators and 128 measures have achieved consensus as being relevant and feasible in general practice according to general practice and PHN staff. This set of indicators and measures has the potential to define high-quality general practice, and inform primary health care quality improvement and funding reforms, aligning with Australia's Primary Health Care 10-Year Plan 2022–2032 to achieve person-centred primary health care and providing the evidence for a VPR platform to incentivise quality person-centred primary health care. However, further consideration about feasibility in the context of systemic, GP-specific and patient-specific barriers, including issues with current health system constraints, existing health data linkage, and general practice buy-in, capacity and time, is required for some of the measures. Future research should include further consultations with rural practitioners; co-design of appropriate patient-reported measures with consumers; and consideration of implementation strategies with general practice stakeholders. Increasing the capacity of primary health care professionals to undertake data collection for quality improvement, including through appropriate support is important for the successful implementation of a tool such as the QUEST PHC. Generalisability of the QUEST PHC tool beyond Australian general practice should also be investigated.

## Supporting information

**S1 File. 79 indicators and their associated 128 measures.**
(PDF)

**S2 File. Practice staff and PHN staff comparison on the feasibility of the indicators and measures.**
(PDF)

**S3 File. Qualitative themes and the associated data.**
(PDF)

## Acknowledgments

We would like to acknowledge the contribution of the participants to this Delphi consensus study, particularly at a very challenging time in Australia and around the world. We also acknowledge the contribution of Dr Sandro Sperandei who provided statistical guidance during data analysis. We acknowledge the Digital Health CRC for its funding support, and the Project Control Group: Digital Health CRC, Brisbane North PHN, Central and Eastern Sydney PHN, Nepean Blue

Mountains PHN, North Western Melbourne PHN, South Western Sydney PHN, Western Sydney PHN (WentWest), Western Australia Primary Health Alliance, and Western NSW PHN; and also the RACGP, ACRRM, Justice Health NSW and SA Prison Health Service for their contribution to the Steering Committee. Digital Health CRC chaired the quarterly Project Control Group meetings during the funding period for administrative and reporting purposes.

## Author contributions

**Conceptualization:** Phyllis Lau, Penelope Abbott, Kathy Tannous, Steven Trankle, Kath Peters, Andrew Page, Tim Usherwood, Jennifer Reath.

**Data curation:** Phyllis Lau, Baneen Alrubayi, Lucy Bannister, Dylan Pakkiam, Natalie Cochrane.

**Formal analysis:** Phyllis Lau, Samantha Ryan, Natalie Cochrane.

**Funding acquisition:** Penelope Abbott, Kathy Tannous, Kath Peters, Andrew Page, Tim Usherwood, Jennifer Reath.

**Investigation:** Phyllis Lau, Penelope Abbott, Kathy Tannous, Steven Trankle, Kath Peters, Andrew Page, Natalie Cochrane, Tim Usherwood, Jennifer Reath.

**Methodology:** Phyllis Lau, Penelope Abbott.

**Project administration:** Phyllis Lau, Samantha Ryan.

**Supervision:** Phyllis Lau.

**Writing – original draft:** Phyllis Lau.

**Writing – review & editing:** Samantha Ryan, Baneen Alrubayi, Lucy Bannister, Dylan Pakkiam, Penelope Abbott, Kathy Tannous, Steven Trankle, Kath Peters, Andrew Page, Natalie Cochrane, Tim Usherwood, Jennifer Reath.

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
