## [Decision Letter · Decision Letter 0]

31 Mar 2025

PONE-D-25-01182Indicators of high-quality general practice to achieve Quality Equity and Systems Transformation in Primary Health Care (QUEST-PHC) in Australia: a Delphi consensus studyPLOS ONE

Dear Dr. Lau,

Thank you for submitting your manuscript to PLOS ONE. After careful consideration, we feel that it has merit but does not fully meet PLOS ONE’s publication criteria as it currently stands. Therefore, we invite you to submit a revised version of the manuscript that addresses the points raised during the review process.

Dear Authors, during the review process, two reviewers indicated their approval for acceptance. However, it has come to attention that the second reviewer, while appearing to support acceptance, actually declined to complete the review. This renders the initial decision of acceptance invalid. Additionally, two other reviewers provided feedback and suggested revisions. Thank you for your attention to this matter.  ==============================

We look forward to receiving your revised manuscript.

Kind regards,

Pengpeng Ye

Academic Editor

PLOS ONE

 [This study was funded by the Digital Health Cooperative Research Centre https://www.digitalhealthcrc.com/.]. 

Additional Editor Comments:

During the review process, two reviewers indicated their approval for acceptance. However, it has come to attention that the second reviewer, while appearing to support acceptance, actually declined to complete the review. This renders the initial decision of acceptance invalid. Additionally, two other reviewers provided feedback and suggested revisions.

Reviewers' comments:

Reviewer's Responses to Questions

**Comments to the Author**

1. Is the manuscript technically sound, and do the data support the conclusions?

Reviewer #1: Yes

Reviewer #2: Yes

Reviewer #3: Partly

Reviewer #4: Yes

2. Has the statistical analysis been performed appropriately and rigorously? 

Reviewer #1: Yes

Reviewer #2: Yes

Reviewer #3: No

Reviewer #4: Yes

3. Have the authors made all data underlying the findings in their manuscript fully available?

Reviewer #1: Yes

Reviewer #2: Yes

Reviewer #3: Yes

Reviewer #4: Yes

4. Is the manuscript presented in an intelligible fashion and written in standard English?

Reviewer #1: Yes

Reviewer #2: Yes

Reviewer #3: Yes

Reviewer #4: Yes

5. Review Comments to the Author

Reviewer #1: The study provides invaluable insights into the critical factors influencing primary healthcare in Australia, particularly in achieving quality, equity, and systems transformation. The comprehensive approach, involving the Delphi consensus method, contributes significantly to the field and serves as an essential resource for those working in primary care and health policy.

General Comments:

• The study mentions achieving "wider consensus," but clarifying who the "wider" audience refers to and whether the findings could be generalized to settings beyond Australian general practice would be beneficial.

Methods Section and Setting:

• I would suggest adding the protocol of this Delphi consensus study (briefly), including the justification of the

methodology.

• It’s beneficial for the readers to add the characteristics of the 162 PHNs, their geographical locations, and the populations in their regions (briefly).

• Please justify the sample size and if any calculations were used to determine it.

• The description of how consensus was determined (≥70% agreement for relevance and feasibility) is good, but providing more context on the threshold chosen for consensus would add clarity. Why was this agreement threshold selected, and is this typical in Delphi studies in healthcare?

Data Collection:

• It would be beneficial if you could clarify if the consent form has been submitted and how and where it was obtained.

The results:

• The results mention that 19 measures were "slightly less feasible" than others. It would be helpful to elaborate on what factors contributed to these measures being rated as less feasible. More specific examples would make the results more actionable for readers.

Conclusions:

• The conclusions are firm but could benefit from further exploration of the practical implications of the results. Specifically, how might the findings influence policy or the future implementation of the QUEST-PHC indicators in general practice? Additionally, while the feasibility of some measures needs further consideration, providing examples or suggestions on how to address these feasibility concerns would strengthen the conclusion.

Reviewer #2: Please stop sending me articles to review.Please stop sending me articles to review.Please stop sending me articles to review.Please stop sending me articles to review.Please stop sending me articles to review.

Reviewer #3: the title the authors and and a statement in the abstract section quite different from that of the concept stated in the title, in my opinion either need a modification. the concurrent mixed approach is purely a quantitative and qualitative method, but you authors did not claim about the quantitative one; this is purely a qualitative so why concurrent mixed methods is applied? also it lacks about the concepts Anonymity, Iteration, Controlled Feedback, and Statistical Stability of Consensus; except a few stated about iteration; even steps in the Delphi process are not well defined. Also there is nothing about indicator development (If I missed ..). it lacks the evidence to set a minimum of the numbers of participants.

Reviewer #4: Thank you for this well written paper that addresses an important area with the potential to advance healthcare equity in primary care. Here are a few of my observations:

The abstract mentioned using thematic analysis, while the main paper also mentioned using content analysis. Could you clarify whether it is thematic analysis? or content analysis was used to gather data into themes? It would be helpful to provide more clarity on this.

If content analysis was used and the qualitative part aimed to identify additional indicators or measures, it would be helpful to specify whether any new indicators or measures were identified. While content analysis doesn’t always require reporting counts, it might be useful to consider quantifying the codes/themes to provide a clearer understanding of how frequently a particular item appears in the data. This could be especially relevant if any recommendations have or will influence the refinement of the indicators or measures. Additionally, a brief statement on how the qualitative findings influenced the indicators or measures, beyond the barriers and facilitators discussed under the discussion section would be helpful.

While the process is well detailed, it would be helpful to indicate the reporting guidelines used for this study.

Best wishes.

6. PLOS authors have the option to publish the peer review history of their article (what does this mean? ). If published, this will include your full peer review and any attached files.

**Do you want your identity to be public for this peer review?** For information about this choice, including consent withdrawal, please see our Privacy Policy .

Reviewer #1: **Yes: ** Sharifa Alblooshi

Reviewer #2: No

Reviewer #3: No

Reviewer #4: No

---

## [Author Response · Author response to Decision Letter 1]

29 May 2025

Please see attachment 'rebuttal letter 20250529'.

---

## [Decision Letter · Decision Letter 1]

17 Jun 2025

Indicators of high-quality general practice to achieve Quality Equity and Systems Transformation in Primary Health Care (QUEST-PHC) in Australia: a Delphi consensus study

PONE-D-25-01182R1

Dear Dr. Lau,

We’re pleased to inform you that your manuscript has been judged scientifically suitable for publication and will be formally accepted for publication once it meets all outstanding technical requirements.

Kind regards,

Pengpeng Ye

Academic Editor

PLOS ONE

Additional Editor Comments (optional):

Reviewers' comments:

Reviewer's Responses to Questions

**Comments to the Author**

1. If the authors have adequately addressed your comments raised in a previous round of review and you feel that this manuscript is now acceptable for publication, you may indicate that here to bypass the “Comments to the Author” section, enter your conflict of interest statement in the “Confidential to Editor” section, and submit your "Accept" recommendation.

Reviewer #1: (No Response)

Reviewer #3: All comments have been addressed

Reviewer #4: (No Response)

2. Is the manuscript technically sound, and do the data support the conclusions?

Reviewer #1: Partly

Reviewer #3: Yes

Reviewer #4: Yes

3. Has the statistical analysis been performed appropriately and rigorously? 

Reviewer #1: Yes

Reviewer #3: Yes

Reviewer #4: Yes

4. Have the authors made all data underlying the findings in their manuscript fully available?

Reviewer #1: Yes

Reviewer #3: Yes

Reviewer #4: Yes

5. Is the manuscript presented in an intelligible fashion and written in standard English?

Reviewer #1: Yes

Reviewer #3: Yes

Reviewer #4: Yes

6. Review Comments to the Author

Reviewer #1: The work is timely and addresses a critical need for standardized metrics to guide quality improvement and health system funding reforms. Thanks for submitting this interesting study!

Reviewer #3: Comments for authors, unless and otherwise I missed it it is good to write the implication section in the article, because by its nature qualitative study needs an implication in terms of practice, methodology and policy.

Reviewer #4: Please move the information on the reporting guidelines from the supplementary information to the Methods section.

Best wishes.

7. PLOS authors have the option to publish the peer review history of their article (what does this mean? ). If published, this will include your full peer review and any attached files.

**Do you want your identity to be public for this peer review?** For information about this choice, including consent withdrawal, please see our Privacy Policy .

Reviewer #1: **Yes: ** Dr Sharifa Alblooshi

Reviewer #3: No

Reviewer #4: **Yes: ** Adeniji Oluwatoyin

---

## [Editor Report · Acceptance letter]

PONE-D-25-01182R1

PLOS ONE

Dear Dr. Lau,

I'm pleased to inform you that your manuscript has been deemed suitable for publication in PLOS ONE. Congratulations! Your manuscript is now being handed over to our production team.

Kind regards,

on behalf of

Dr. Pengpeng Ye

Academic Editor

PLOS ONE